# Ursolic Acid Impairs Cellular Lipid Homeostasis and Lysosomal Membrane Integrity in Breast Carcinoma Cells

**DOI:** 10.3390/cells11244079

**Published:** 2022-12-16

**Authors:** Ditte L. Fogde, Cristina P. R. Xavier, Kristina Balnytė, Lya K. K. Holland, Kamilla Stahl-Meyer, Christoffel Dinant, Elisabeth Corcelle-Termeau, Cristina Pereira-Wilson, Kenji Maeda, Marja Jäättelä

**Affiliations:** 1Cell Death and Metabolism Group, Center for Autophagy, Recycling and Disease (CARD), Danish Cancer Society Research Center, 2100 Copenhagen, Denmark; 2Department of Biology, University of Minho, 4710-057 Braga, Portugal; 3Core Facility for Bioimaging, Danish Cancer Society Research Center, 2100 Copenhagen, Denmark; 4Centre of Biological Engineering, LABBELS Associate Laboratory, University of Minho, 4710-057 Braga, Portugal; 5Department of Cellular and Molecular Medicine, Faculty of Health Sciences, University of Copenhagen, 2200 Copenhagen, Denmark

**Keywords:** autophagy, cationic amphiphilic drugs, cell death, cancer, lysosomal membrane permeabilization, ursolic acid

## Abstract

Cancer is one of the leading causes of death worldwide, thus the search for new cancer therapies is of utmost importance. Ursolic acid is a naturally occurring pentacyclic triterpene with a wide range of pharmacological activities including anti-inflammatory and anti-neoplastic effects. The latter has been assigned to its ability to promote apoptosis and inhibit cancer cell proliferation by poorly defined mechanisms. In this report, we identify lysosomes as the essential targets of the anti-cancer activity of ursolic acid. The treatment of MCF7 breast cancer cells with ursolic acid elevates lysosomal pH, alters the cellular lipid profile, and causes lysosomal membrane permeabilization and leakage of lysosomal enzymes into the cytosol. Lysosomal membrane permeabilization precedes the essential hallmarks of apoptosis placing it as an initial event in the cascade of effects induced by ursolic acid. The disruption of the lysosomal function impairs the autophagic pathway and likely partakes in the mechanism by which ursolic acid kills cancer cells. Furthermore, we find that combining treatment with ursolic acid and cationic amphiphilic drugs can significantly enhance the degree of lysosomal membrane permeabilization and cell death in breast cancer cells.

## 1. Introduction

Ursolic acid (3β-hydroxy-urs-12-en-28-oic acid, UA) is a naturally occurring pentacyclic triterpenoid. It is abundantly present in various fruits and vegetables, especially in rosemary, thyme, and the peel of apples [1]. Akin to many other bioactive pentacyclic triterpenoids, UA has been suggested to be responsible for the pharmacological effects of various medicinal plants [2], such as *Oldenlandia diffusa*, which is a commonly prescribed remedy against cancer in traditional Chinese medicine [3]. To our knowledge, no results from clinical trials of UA have yet been reported. Nevertheless, UA has been intensively investigated for its various beneficial effects on human health, including its anti-inflammatory, anti-obesity, anti-oxidative, anti-diabetic, cardioprotective, neuroprotective, hepatoprotective and anti-cancer activities [4]. Its anti-cancer activity has been mainly connected to its ability to cause apoptotic cell death, for example by upregulating the expression of cellular tumor antigen p53, downregulating the expression of apoptosis regulator Bcl-2 or by activating Wingless/Integrated (Wnt)/β-catenin signaling [5,6]. The ability of UA to induce cancer cell apoptosis by modulating these and other signaling pathways has been confirmed by several studies [4]. Some recent reports suggest, however, that UA may also activate non-apoptotic cell death pathways associated with autophagosome and lysosome accumulation [7,8,9]. In addition to cell death induction, UA can limit cancer growth through its anti-proliferative and anti-invasive effects, for example, by inhibiting the cancer-promoting transcription factors, Nuclear Factor-κB (NF-κB) [10,11] and Signal Transducer and Activator of Transcription 3 (STAT3) [12], or the mammalian Target Of Rapamycin (mTOR) signaling pathway [13]. Moreover, UA reverts multi-drug resistance in various cancer cell lines [14,15].

As mentioned above, UA-induced non-apoptotic cell death has been associated with an accumulation of autophagosomes and lysosomes [7,8,9]. A recent study on cell death induced by UA232, a UA derivative with significantly enhanced anticancer activity, suggests that UA232-induced apoptosis is associated with endoplasmic reticulum stress and lysosomal dysfunction [16]. The latter is of special interest in cancer treatment because cancer cells are highly reliant on intact lysosomal recycling pathways, endocytosis and autophagy, to supply the energy and building materials for their rapid growth [17]. Furthermore, cancer progression is associated with decreased lysosomal membrane integrity [18], and the disruption of lysosomal membranes in cancer cells can induce cell death by activating apoptosis, necrosis, or lysosome-dependent cell death pathways depending on the degree of lysosomal damage and the cell type in question [19,20]. Notably, lysosome-dependent cell death can be induced even in cancer cells highly resistant to commonly used apoptosis-inducing cancer drugs [11,21,22,23]. Cationic amphiphilic drugs (CADs) are examples of drugs that target cancer cell lysosomes [24]. Due to their amphiphilic and basic nature, they diffuse rapidly into lysosomes and are trapped there following the protonation of a tertiary amine group at the lysosomal pH of ~4.5 [25,26]. Their accumulation in the lysosomal lumen neutralizes the lysosomal pH and the negative surface charge of intraluminal lysosomal vesicles, leading to the inhibition of luminal acid sphingomyelinase (aSMase/*SMPD1*) and other luminal hydrolases [27,28]. The resultant accumulation of sphingomyelin (SM) and lysoglycerophospholipids (lysoGPLs) promotes lysosomal membrane permeabilization (LMP), which leads to the release of cathepsins and other cytotoxic lysosomal contents into the cytosol, where they trigger lysosome-dependent cell death [24,29,30,31,32].

As an acid (p*K*_a_ = 4.73) without an amine group (Figure 1A), UA is structurally and physicochemically distinct from lysosomotropic CADs, which are typically weak bases with p*K*_a_ > 6.5 [25]. In this report, we present data showing that UA, nevertheless, targets the lysosomal pathways in MCF7 breast cancer cells. Prompted by this observation, we studied the effects of UA in further detail focusing on the pH, hydrolytic activity and membrane integrity of lysosomes, the maturation of autophagosomes, and cellular lipid homeostasis. Our data demonstrate that despite its clearly distinct physicochemical properties, UA in large part shares its effects on these cellular parameters with CADs. Supporting their distinct but similar mechanisms of action, the simultaneous targeting of lysosomes with UA and ebastine, a CAD antihistamine, resulted in a synergistic induction of LMP and cell death. These data encourage further studies on the anticancer activity of UA in combination with CADs and other lysosome-targeting drugs.

## 2. Materials and Methods

### 2.1. Materials

For a detailed list of all the reagents, see Appendix A. All drugs were dissolved in DMSO except for the TNF-α which was dissolved in sterile water.

### 2.2. Cell Culture and Treatments

The MCF7 cell line used was a TNF-sensitive subclone of MCF7 human breast carcinoma cells (MCF7-S1) [33]. The MCF7-S1 was cultured in Roswell Park Memorial Institute 1640 medium (RPMI-1640) containing GlutaMAX (Thermo Fisher Scientific, Waltham, MA, USA) supplemented with 6% (*v*/*v*) fetal calf serum (FCS, Life Technologies, Carlsbad, CA, USA), and 1/4X penicillin/streptomycin (Life Technologies). Its transfected derivatives, MCF7-eGFP-LC3, MCF7-eGFP-LC3(G120A), MCF7-pCEP, MCF7-BCL2 [34], MCF7-tfLC3 [35], MCF7-RLuc-LC3, MCF7-RLuc-LC3(G120A) [36] and MCF7-eGFP-LGALS3 [37] have been described previously. HCT15 human colorectal carcinoma cells from American Type Culture Collection (ATCC, Danvers, MA, USA) were cultured in RPMI-1640 containing GlutaMAX supplemented with 6% (*v*/*v*) FCS, and 1/4X penicillin/streptomycin. The HCT116 human colorectal carcinoma cells provided by Dr. S. Shirasawa (Fukuoka University, Fukuoka, Japan) [38] and HeLa human cervix carcinoma cells were cultured in Dulbecco’s modified Eagle’s medium (DMEM) containing GlutaMAX (Thermo Fisher Scientific) with 10% (*v*/*v*) FCS and 1/4X penicillin/streptomycin.

### 2.3. Cell Death Assays

Cell death was measured either after propidium iodide (PI, Sigma-Aldrich, St. Louis, MO, USA) staining or by a lactate dehydrogenase (LDH) release assay (Roche, Basel, Switzerland). For the PI staining, the cells were collected (both floating and attached cells) and washed in a buffer containing ice cold Dulbecco’s phosphate buffered saline (DPBS, Thermo Fisher Scientific) with 5% (*v*/*v*) FCS. The cells were then resuspended in the same buffer and PI added to a final concentration of 0.5 mg/mL. The samples were kept on ice and protected from light until analysis. The stained suspensions (20 µL) were placed on microscope slides, overlaid with coverslips and images were acquired on a fluorescent microscope. The percentage of dead cells (PI positive) was calculated from the number of PI positive cells and total number of cells (visualized under a phase contrast), using at least 500 cells per slide.

For the LDH release assay, 6000 cells per well in 200 μL of colorless RPMI medium 1640 containing GlutaMAX (Thermo Fisher Scientific) supplemented with 6% (*v*/*v*) FCS + 1/4X Penicillin/Streptomycin were seeded on a 96-well plate. After the indicated treatments, 50 μL of the medium was transferred to a new plate, the remaining medium was removed, and the cells were lyzed for 30 min at 37 °C in 200 μL of medium containing 1% (*v*/*v*) Triton X-100 (Sigma-Aldrich). The cell lysates (50 μL) were transferred to a new plate, and the LDH activity in the media and cell lysates was measured by adding 50 μL of the dye and catalyst mixture from Roche (45:1 (*v*/*v*) ratio for mixing). The plates were incubated in the dark on a rocking table for 10–30 min and the absorbance was measured on a Varioskan plate reader (Thermo Fisher Scientific) at 490 and 690 nm (background). The value of a blank well was subtracted. The cell death was calculated as following:$$\mathrm{cytotoxicity}\left(\%\right)=\frac{\mathrm{absorbance}\left[\mathrm{medium}\right]*100}{\mathrm{absorbance}\left[\mathrm{medium}\right]+\mathrm{absorbance}\left[\mathrm{lysate}\right]}$$

### 2.4. Luciferase-Based Autophagic Flux Assay

The autophagic flux was analyzed using MCF7 cells expressing Renilla reinformis luciferase fused with either wild type LC3 (MCF7-RLuc-LC3) or an autophagy-defective mutant of LC3 (MCF7-RLuc-LC3(G120A) as described previously [36]. For the luciferase-based real-time assay in living cells, the cells were plated at 8 × 10^4^ cells/mL for 24 h. Afterwards, the cells were incubated for 2 h in 60 μL of medium containing 50 nM of EnduRen^TM^ (Promega, Madison, WI, USA) and the luminescence was measured (Enspire 2300 Multilabel reader, Perkin Elmer, Waltham, MA, USA). UA (Abcam, Cambridge, UK), rapamycin (Sigma-Aldrich) or both dissolved in an EnduRen^TM^ containing medium were added to the cells and the luminescence measurements performed every second hour for 20 h. The ratio in luminescence between the RLucLC3 and RLucLC3(G120A) was calculated and normalized to time 0.

### 2.5. Immunocytochemistry

The cells (5 × 10^4^) were cultured in a 24-well plate with one glass coverslip in each well. The cells were fixed by incubating with 4% (*v*/*v*) paraformaldehyde in PBS (Ampliqon. Odense, Denmark) for 8 min at 37 °C, washed twice in DPBS, and subsequently treated with NH_4_Cl (50 mM) (Sigma-Aldrich) for 10 min. The cells were washed twice in DPBS and permeabilized using 100% methanol (VWR, Radnor, PA, USA). The cells were washed using DPBS and blocked using buffer 1 (DPBS containing 1% bovine serum albumin (BSA) (VWR) (*w*/*v*) and 0.3% (*v*/*v*) Triton-X-100) and 5% goat serum (*v*/*v*) for 20 min. Subsequently, the coverslips were incubated with the primary antibodies in buffer 1 with 5% (*v*/*v*) goat serum (DAKO, Carpinteria, CA, USA) for 18 h. The coverslips were washed three times in buffer 1 and subsequently incubated in buffer 2 (DPBS containing 0.25% (*w*/*v*) BSA and 0.1% (*v*/*v*) Triton-X-100) with secondary antibodies at room temperature for 1 h. The coverslips were washed three times for 5 min using buffer 3 (DPBS containing 0.05% (*v*/*v*) Tween-20 (VWR)). The nuclei were stained with 2.5 μg/mL of Hoechst 33342 (Sigma-Aldrich). The coverslips were washed three times in DPBS and subsequently mounted on glass slides using a Prolong Gold Antifade mounting medium (Life Technologies).

### 2.6. LC3-Puncta Formation Assays

The number of LC3-puncta positive cells was assessed in fixed MCF7-eGFP-LC3 and MCF7-eGFP-LC3(G120A) cells by counting the percentages of cells with the indicated number of LC3-puncta (a minimum of 2 × 100 cells/sample) using a Zeiss510 Confocal Laser Scanning Microscope (Carl Zeiss, Stuttgart, Germany). In the HCT15 cells, LC3-puncta formation was analyzed after immunostaining endogenous LC3. The fluorescent signal from each puncta in a MCF7-tfLC3 cell was quantified using ImageJ (Java 1.8). The ratio of the pH-sensitive GFP signal divided by the pH-insensitive RFP signal was calculated. Each cell was counted as a separate point. A total of five cells were quantified per repetition.

### 2.7. Measuring Nuclear Proximity of LAMP2 and EEA1

The cells were imaged with a 20X objective on an Olympus ScanR screening microscope (Olympus, Tokyo, Japan) using the standard DAPI (chromatin), Cy3 (for EEA1) and Cy5 (for LAMP2) filters. The image processing and analyses were performed with the ScanR analysis software (Olympus). Nuclei were segmented by intensity thresholding the DAPI signal. This nuclear segmentation mask was modified to form four touching concentric rings, the smallest of which had an internal edge starting three pixels inside the original nuclear mask and with all rings having a width of six pixels (~2 µm). The mean intensity of the EEA1 and LAMP2 signals inside these rings was measured. The nuclear proximity was calculated by the ratio of the mean intensity in ring 1 divided by the average of mean intensities in rings 2, 3, and 4. Data was visualized as SuperPlot [39].

### 2.8. Lysosomal pH Measurements

Cells (1.5 × 10^4^) were seeded in an 8-well slide and cultured for 24 h before treatment with 1.25 mg/mL fluorescein (FITC)- and tetramethylrhodamine (TMR)-coupled dextran for approximately 16 h and washed twice in DPBS before adding regular medium for a chase period of 3 h. The treatment of cells was performed during the chase period. At the end of the chase period/treatment, 2.5 mg/mL Hoechst 33342 (Sigma-Aldrich) (1:4000) was added and the cells were washed with DPBS after 5 min. Imaging solution was added and images were acquired using an LSM800 confocal microscope and the ZEN 2010 software (Carl Zeiss). The intensity of the FITC and TMR signals were quantified using ImageJ and the ratio was calculated.

### 2.9. Cystein Cathepsin and NAG Activities

The cysteine cathepsin (zFR-AFC) and *N*-acetyl-*β*-glucosaminidase (NAG) activities were measured as described previously [40]. Briefly, the cytosolic fraction was extracted with 20 μg/mL digitonin and the total cellular fraction with 200 μg/mL digitonin (Sigma-Aldrich). The rate of the appropriate substrate hydrolysis (Vmax) was measured over 20 min at 30 °C on a SpectraMax Gemini fluorometer (Molecular Devices, San Jose, CA, USA). The obtained values were divided by the LDH activity (determined by the Roche cytotoxicity detection kit) for normalization.

### 2.10. Protein Concentration Measurements

The protein concentration was measured using a Pierce^TM^ bicinchoninic acid (BCA, Thermo Fisher Scientific) assay kit according to the manufactures protocol. The protein concentrations were calculated from a BSA standard curve.

### 2.11. Western Blotting

Cells were harvested in a Radio-ImmunoPrecipitation assay (RIPA) lysis buffer (TRIS 50 mM, NaCl 150 mM, EDTA 1 mM, Triton X 1% (*v*/*v*), sodium deoxycholate 1% (*w*/*v*), and SDS 0.1% (*w*/*v*)) and mixed with a 2X Laemmli sample buffer (LSB) (Tris 0.125 M, glycerol 20% (*v*/*v*), bromophenol blue 0.4 g/L, and SDS 0.14 M) + 100 mM DTT and heated to 95 °C for 5 min before running on a 15-well MiniProtean TGX gels (4–15%) (Bio-Rad, Hercules, CA, USA). The proteins were transferred to a nitrocellulose membrane with a Bio-Rad trans-blot turbo system. The membrane was blocked using PBS-T (1X PBS and 0.1% Tween-20 (*v*/*v*)) with 5% BSA for 45 min and subsequently incubated in PBS-T with 5% BSA and the primary antibodies of interest for 18 h at 5 °C. The membranes were washed in PBS-T and incubated with PBS-T with 5% BSA and the secondary antibodies for 1 h. The membrane was washed in PBS-T and subsequently incubated with Clarity Western Enhanced Chemiluminiscent (ECL) reagents (Bio-Rad) or a SuperSignal West Femto Maximum Sensitivity Substrate (Thermo Fisher Scientific). The images were acquired using Fujifilm LAS-4000 (Tokyo, Japan).

### 2.12. Galectin-3 Puncta Assay

The galectin puncta assay was used to visualize LMP as described previously [37,41]. Subconfluent MCF7 cells stably expressing eGFP-tagged galectin-3 (MCF7-eGFP-LGALS3), grown in Greiner 96-well plates (Greiner Bio-One) were treated with 0.1 μM of Nuclear Violet (AAT Bioquest, Pleasanton, CA, USA) and 0.2 μM of SiR-Tubulin (Spirochrome, Thurgau, Switzerland) 2 h prior to the imaging using the ImageXpress Micro Confocal system (Molecular Devices, San Jose, CA, USA) and MetaXpress analysis software (Molecular Devices) according to the manufacturer’s instructions. In short, the images were acquired in widefield mode using a 60X air objective. The image analysis was performed by the MetaXpress analysis software using an analysis pipeline created in the custom module. The images were exposed to cell segmentation, where the nuclei (Nuclear Violet) and cytoplasm (SiR-Tubulin) were masked based on fluorescent intensities and sizes to define individual cells. Border objects were removed to analyze the whole cells. The galectin-3 puncta were counted by detecting punctate objects based on the fluorescent intensity and pixel size inside a defined cell. A “top hat” module was added to detect true puncta and to exclude artefacts arising from a fluorescence accumulation at the edge of the cells. The analysis output was the total cell number, cells positive for galectin-3 puncta (defined as a cell with at least three galectin-3 puncta), and the number of galectin-3 puncta in each galectin-3 puncta positive cell.

### 2.13. Lipid Extraction and Lipidomics

The harvesting of samples and lipid extraction were performed as described by Nielsen et al. [42]. In brief, ~10^5^ cells were added to 1 mL of chloroform/methanol 2:1 (*v*/*v*) in addition to 10 μL of internal lipid standard mix and 20 μL of internal standard for ebastine/carebastine 1:1 in methanol (see Appendix A for a list of the internal lipid standards used in this study). The lipid extraction was performed with a modified Bligh and Dyer protocol [43]. Samples were shaken at 2000× *g* rpm for 15 min at 4 °C and subsequently centrifuged for 3 min at 7800× *g* and 4 °C, after which the lower phase was transferred to a new tube. The solvent was evaporated, and the samples were dissolved in chloroform/methanol 1:2 (*v*/*v*). The samples were mixed with either positive (13.3 mM of ammonium bicarbonate in isopropanol) or negative ionization solvents (0.2% (*v*/*v*) tri-ethyl-amine in chloroform/methanol 1:5 (*v*/*v*)). The samples were infused and analyzed in a positive and negative mode using a quadrupole-Orbitrap mass spectrometer Q Exactive (Thermo Fisher Scientific) equipped with a TriVersa NanoMate (Advion Biosciences, Ithaca, NY, USA). The mass spectra were analyzed using LipidXplorer [44], a python-based software. The absolute quantities were calculated using LipidQ, which is an in-house built R-based software (https:/github.com/ELELAB/lipidQ, accessed on 3 March 2021). The absolute molar quantities were determined based on the intensity value of the lipids compared to their internal lipid standard. The ebastine uptake in the cells was quantified by comparing the intensity value of ebastine to the value of its internal standard and normalizing to the total lipid content in the sample (see Appendix A for a list of the internal lipid standards used in this study).

### 2.14. Lipidomics Data Analysis

The lipid nomenclature is as described previously [45]. To analyze the data, the R statistical software (R Core Team (2021)) version 1.4.1717 was used. Lipids with a median quantity of all the replicates within a sample type of less than 0.0001 mol% were excluded from the analysis. For a statistical analysis of every lipid species, class or category, the “limma” package [46] was used to fit a linear model based on the lipid quantity in the samples. Log2-transformed fold change values and the associated *p*-values were calculated. To correct for multiple testing, the Benjamini–Hochberg method was used. Lipids were considered significantly changed if the adjusted *p*-value (*q*-value) was below 0.05. Heatmaps were made using the “pheatmap” package [45].

### 2.15. Statistical Analysis

Statistics were performed using the RStudio (version 2021.9.1.372), R and GraphPad Prism software 9.1.0 on the lipidomics data (see Section 2.14). For all data besides the lipidomics, the GraphPad Prism software 9.1.0 was used. The data are shown as a mean ± standard deviation (SD) of at least three independent experiments unless otherwise stated in the figure legend. All the statistics were performed on the mean values from at least three independent experiments unless otherwise stated in the figure legend. All the *t*-tests were two-tailed. The statistical tests used are specified in the figure legends. The *p*-values are shown for every significant comparison.

## 3. Results

### 3.1. UA Kills MCF Breast Cancer Cells Partly via Apoptosis

To enlighten the molecular mechanisms of the cytotoxicity of UA (Figure 1A) on cancer cells, we first examined its ability to kill various types of cancer cells in culture by detecting plasma membrane-permeabilized cells using a membrane-impermeable propidium iodide (PI) dye. In agreement with previous reports [7,9,47,48], treatment with UA for 24 h induced death in human breast adenocarcinoma (MCF7) cells, osteosarcoma (U2OS) cells, cervical adenocarcinoma (HeLa) cells, and colorectal carcinoma (HCT116) cells in a dose-dependent manner, with lethal dose 50 (LD_50_) values of 12.8 µM, 7.7 µM, 16.8 µM, and 19 µM, respectively (Figure 1B; Appendix A). Even at a concentration of 8 μM that only killed ~5.5% of MCF7 cells in 24 h (Figure 1B), a treatment for 48 h reduced the total cell count of the MCF7 cells with >50% in comparison to the untreated or vehicle-treated cells (Figure 1B), suggesting that UA inhibited the cell proliferation. Next, we investigated the kinetics and dose response of UA-induced cytotoxicity in the MCF7 cells using a lactate dehydrogenase (LDH) release assay. While the treatment with UA at concentrations up to 8 µM for up to 48 h failed to induce a significant cell death, treatments with 12 and 16 µM of UA for 48 h killed 38.8% and 67.9% of the MCF7 cells, respectively (Figure 1C). The cells treated with 16 µM UA started dying already after 4 h of treatment but the effect remained statistically insignificant until 16 h (Figure 1C). It should be noted here, that despite the careful handling of the drug, different batches of UA showed some variation in their killing efficacies.

Even though UA induces apoptosis in various cancer cell lines [4], most recent studies have reported that apoptosis only accounts for a part of the mechanism of UA-induced cell death [7,8]. To examine whether UA induces apoptosis in MCF7 cells, we next measured the UA-induced cell death in MCF7 cells transfected with an expression vector encoding an anti-apoptotic protein B-cell lymphoma 2 (Bcl2) or an empty control vector pCEP (Figure 1D). As expected, a Bcl2 overexpression almost completely inhibited the cell death induced by an apoptosis inducer, namely, Tumor Necrosis Factor α (TNF-α), while it did not affect the cell death induced by siramesine, a cationic amphiphilic drug (CAD) that induces lysosome-dependent cell death in MCF7 cells [49,50] (Figure 1D). In the case of the UA-treated cells, the Bcl2 overexpression provided a partial protection against UA (Figure 1D; Appendix A). Taken together, these data indicate that apoptosis plays only a minor role in the UA-induced cytotoxicity in MCF7 cells.

### 3.2. UA Inhibits Autophagosome Maturation

Since UA-induced cell death has previously been linked to the activation of autophagy [7,9], we next investigated the ability of UA to induce autophagic flux in MCF7 cells. For this purpose, we monitored the microtubule-associated protein 1 light chain 3 (LC3) turnover using a luciferase-based assay previously developed in our laboratory [36]. This assay monitors the autophagic flux by comparing the levels of Renilla Luciferase (RLuc)-tagged wild type LC3 (RLuc-LC3) and a similarly tagged autophagy-deficient LC3(G120A) mutant (RLuc-LC3(G120A)). The RLuc-LC3/RLuc-LC3(G120A) ratio inversely reflects autophagic flux, as only the RLuc-LC3 is degraded by autophagy [51]. Twenty-hour treatment of the reporter-expressing cells with a sublethal concentration (8 µM) of UA increased this ratio by 25.4%, as was previously observed when treating with autophagy inhibitors [36]; however, it should be noted that this effect of UA was not significant. A similar treatment with an autophagy inducer rapamycin reduced the ratio by over 50% (Figure 2A). Furthermore, a co-treatment with UA completely inhibited the rapamycin-induced decrease in the RLuc-LC3/RLuc-LC3(G120A) ratio (Figure 2A). These data indicate that UA effectively inhibits autophagic flux in MCF7 cells.

To investigate at which stage of the autophagic process UA blocks the flux, we investigated the LC3 puncta formation in the MCF7 cells stably expressing tandem fluorescent LC3 (tfLC3), i.e., LC3 fused to pH-sensitive, enhanced green fluorescent protein (eGFP) and pH-insensitive monomeric red fluorescent protein (mRFP) [35]. In these cells, the initial autophagic structures such as phagophores and autophagosomes appear as yellow cytoplasmic puncta owing to a fluorescence emission from both fluorophores, whereas mature autolysosomes appear as red puncta due to the quenching of the eGFP signal at the acidic pH (Figure 2B) [53]. Only occasional yellow puncta were visible in the untreated MCF7-tfLC3 cells (Figure 2C). In line with the induction of autophagic flux, the treatment with rapamycin for 24 h induced the appearance of predominantly red puncta. After the treatment with UA, we observed an accumulation of yellow puncta, which could be indicative of either an induced autophagic flux or an inhibition of a late step in the autophagic pathway (Figure 2B,C). The eGFP/mRFP intensity ratios quantified for the puncta in the UA-treated cells were indeed higher than those in the rapamycin-treated cells, indicating that the UA-treated cells accumulated initial phagophores/autophagosomes (Figure 2D). Furthermore, the cells treated with rapamycin had a lower background signal from cytosolic tfLC3 compared to the UA- or vehicle-treated cells indicating that autophagy was ongoing and the tfLC3 was continuously broken down. Overall, this data suggests that UA either inhibits the fusion of mature autophagosomes with lysosomes or causes the formation of non-acidic autolysosomes. The images of cells stained for LAMP2 and LC3 revealed low levels of overlap in the signal, giving the impression that fusion could be impaired after an UA treatment (Appendix A). Supporting the autophagic nature of the puncta observed in the UA-treated MCF7-tfLC3 cells, UA induced an accumulation of green puncta in the MCF7 cells expressing wild type eGFP-LC3 but not in the cells with the autophagy-defective eGFP-LC3(G120A) mutant (Appendix A). Immunostaining of HCT-15 colorectal cancer cells for LC3 revealed that an UA-induced LC3 puncta formation also occurred in these cells (Appendix A), indicating that the effect of UA on autophagy is not limited to MCF7 cells.

### 3.3. UA Alters Lysosomal Trafficking and pH

The final step in the maturation of autophagosomes is their fusion with lysosomes and the formation of autolysosomes, where the degradation of the cargo occurs [54]. Therefore, we examined whether UA inhibits the autophagosome maturation process by impairing the critical functions of lysosomes. To this end, we first visualized the lysosomes in MCF7 cells using an antibody against Lysosome-Associated Membrane Protein 2 (LAMP2), a transmembrane protein residing in the limiting membrane of lysosomes [55]. Whereas numerous LAMP2-positive puncta were dispersed throughout the cytoplasm of the untreated MCF7 cells, treatment with 8–12 μM of UA for 6 h induced their polarized clustering at the juxtanuclear sites (Figure 3A,B). Staining with an antibody against Early Endosome Antigen 1 (EEA1) revealed a similar clustering of early endosomes (Figure 3A,C), although this was not significant. These data suggest that UA regulates the cellular trafficking of endolysosomal vesicles, possibly by interfering with their microtubular trafficking mechanisms. UA likewise translocated lysosomes in HeLa cells, but towards the cell periphery (Appendix A). 

The observed UA-induced juxtanuclear clustering of the lysosomes was reminiscent of the phenotype of leaky lysosomes in MCF7 cells treated with lysosomotropic CADs [50,56]. Encouraged by this similarity, we investigated the effects of UA on the lysosomal function and integrity. First, we monitored changes in the lysosomal pH in the UA-treated MCF7 cells, whose lysosomes were loaded with dextran conjugated to pH-sensitive fluorescein thiocyanate (FITC) and pH-insensitive tetramethylrhodamine (TMR). In agreement with the FITC being quenched in acidic lysosomes, a two hour treatment of the MCF7 cells with a vacuolar-type H^+^-ATPase inhibitor, concanamycin A (ConA), increased the mean FITC/TMR intensity ratio over eight-fold (Figure 3D). The treatment of MCF7 cells with 10 μM of UA for 1–3 h elevated the mean FITC/TMR ratio gradually reaching statistical significance, over a three-fold increase by the end of the treatment (Figure 3D). Thus, UA elevated the lysosomal pH several hours before the loss of plasma membrane integrity (Figure 1C and Figure 3D). 

### 3.4. UA Causes Lysosomal Membrane Permeabilization Prior to BAX Activation

The trafficking of most proenzymes of lysosomal hydrolases to the lysosomes and their maturation from proenzymes to active enzymes depend on the acidic lysosomal pH [57]. In line with the UA-induced increase in lysosomal pH (Figure 3D), this inhibited the activities of lysosomal cysteine cathepsin B and L (zFRase) and *N*-acetyl-*β*-glucosaminidase (NAG) in the MCF7 cells almost as effectively as the ConA (Figure 4A). Immunoblotting of the proteins from the MCF7 cells treated with 8 μM of UA for 2–24 h revealed a clear reduction in the mature cathepsin B (CTSB) levels already after 8 h (Figure 4B).

To investigate whether the UA-induced elevation in lysosomal pH and reduction in lysosomal hydrolase activity were associated with the leakage of lysosomal hydrolases to the cytosol, we measured the hydrolase activities in the cytosolic fractions and total cell lysates of the MCF7 cells left untreated or treated with 8 μM of UA. The UA treatment elevated the cytosolic NAG and zFRase activity from 1.9% and 2% of the total activity in the control cells to 7.4% and 5.3%, respectively. This indicates that UA induced the cytosolic leakage of NAG and zFRase, even though this was not statistically supported (Figure 4C). To further challenge the ability of UA to induce the permeability of lysosomal membranes to lysosomal hydrolases and other proteins, we took advantage of a cytosolic, β-galactoside-binding lectin, galectin-3 (gene = *LGALS3*). Upon LMP, galectin-3 binds with high affinity to the β-galactoside-rich lysosomal glycocalyx, thereby marking leaky lysosomes [37]. Treatment of the MCF7 cells stably expressing eGFP-tagged galectin-3 (MCF7-eGFP-LGALS3) with 8 or 16 μM of UA for 24 h increased both the percentage of eGFP-galectin-3 puncta-positive cells and the number of eGFP-galectin-3 puncta per positive cell in a time-dependent manner (Figure 4D,E). Importantly, the endogenous galectin-3 also formed puncta in the UA-treated MCF7 cells and these puncta co-localized with the lysosomal LAMP2 (Figure 4F), strongly suggesting that the galectin-3 puncta represented leaky lysosomes. The data presented above suggest that the lysosomal integrity was compromised already within the first hours of treatment of the MCF7 cells with UA (Figure 4D,E) and thatLMP was a relatively direct consequence of the UA treatment. We have earlier shown that in MCF7 cells treated with TNF-α, LMP is induced downstream of the mitochondrial outer membrane permeabilization (MOMP), an event of a “point of no return” in the intrinsic apoptosis pathway. Vice versa, LMP occurs upstream of MOMP in MCF7 cells treated with lysosomotropic drugs such as L-leucyl-L-leucine methyl ester (LLOMe) and CADs [37]. To define the order of LMP and MOMP in UA-treated cells, we immuno-stained them for galectin-3 and active Bcl2-associated X protein (BAX), a marker of MOMP. As expected, the treatment with TNF-α for 24 h rendered 58.7% and 2.2% of MCF7 cells single positive for BAX puncta and double positive for both BAX and galectin-3 puncta, respectively, while cells positive for only galectin-3 puncta were virtually absent (Figure 4G,H). On the other hand, treatment with 8 μM of UA for 6 h resembled that of LLOMe and rendered 54% and 11.5% of the MCF7 cells single positive for galectin-3 puncta and double positive for BAX and galectin-3 puncta, respectively, while virtually no cells positive only for BAX puncta were observed (Figure 4G,H). Increasing the UA concentration to 16 µM increased the number of double-positive cells but failed to induce any cells with only BAX puncta (Figure 4G,H). Taken together, these data indicate that akin to lysosomotropic drugs, UA induces the leakage of lysosomes clearly upstream of MOMP.

### 3.5. UA Induces Changes in the Lipid Profile of MCF7 Cells

Lysosomotropic CADs dysregulate lysosomal lipid metabolism through the inhibition of luminal lipid hydrolases, including acid sphingomyelinase (aSMase, *SMPD1*), which degrades sphingomyelin (SM), the most abundant lipid class in the sphingolipid (SL) category, to ceramide (Cer) [24]. This property of CADs is responsible for phospholipidosis, a relatively benign side effect frequently observed among long-term CAD users [58]. This side effect can, however, be exploited in cancer therapy since CAD-induced changes in lysosomal lipid catabolism destabilize cancer cell lysosomes and sensitize cancer cells to lysosome-dependent cell death [24,31,58]. Prompted by similarities between the UA- and CAD-induced lysosomal alterations, we next investigated whether UA also dysregulated the lipid metabolism. To this end, we applied an in-house platform of mass spectrometry (MS)-based shotgun lipidomics, and quantified the lipid species of 22 classes in MCF7 cells treated with the vehicle only, a CAD ebastine (5 µM, for 6 h) or the UA (8 µM, for 8 h) (see Appendix A for the ions used for the lipid identification). We identified an average of 292.8 ± 38.7 lipid species in the control samples, 294.8 ± 37.0 species in the ebastine-treated samples, and 298.2 ± 41.2 in the UA-treated samples (in total, 339, 340 and 341 lipid species, respectively). Each of the 22 lipid classes in the produced dataset, thus, typically included multiple species with minor structural variations, e.g., fatty acids with different numbers of carbon atoms and double bonds. The determined absolute molar quantities of individual lipid species were expressed in a molar percentage (mol%) by normalizing to the total molar quantities of all the monitored lipid species in the same sample.

The treatment with UA or ebastine statistically significantly altered the levels of 132 and 59 lipid species in the MCF7 cells, respectively, when compared to the vehicle-treated cells (Figure 5A), indicating that treatment with UA dysregulates the lipid metabolism of MCF7 cells. UA and ebastine altered the levels of the same 27 lipid species (Figure 5B), including five Cer and seven SM species. In addition, both treatments affected the species in a broad range of lipid classes (Figure 5C). Akin to ebastine, the UA treatment downregulated all the significantly changed Cer species except one, while it upregulated all the significantly changed SM species (Figure 5D), in agreement with the inhibition of aSMase [24]. UA slightly downregulated the entire Cer lipid class and slightly upregulated the entire SM lipid class, (Figure 5E,F). In addition, UA increased the levels of multiple lysoglycerophospholipid (lysoGPL) classes (Figure 5F), in a manner similar to the previously reported effect of CADs in leukemia cells [31]. LysoGPLs are minor GPLs with the presence of only one acyl (alkyl) chain. LysoGPLs have detergent-like properties on lipid membranes, thus their intracellular levels are carefully regulated [59]. UA increased the total lysoGPL level from 0.55 ± 0.13 mol% in the control to 1.06 ± 0.27 mol% (Figure 5F), while ebastine caused a similar effect although to a smaller extent (Figure 5F). See Appendix A for more information on the changes in the lipid profile.

The treatment with UA additionally altered the levels of numerous species of diacylGPL classes (Figure 5C) such as phosphatidylcholine (PC) and phoshatidylethanolamine (PE), without notably altering the levels of these classes (Figure 5E). These alterations led, however, to a reduction in the average number of total acyl double bonds within the diacylGPLs (Figure 5G,H), a change that was also evident for lysoGPLs (Figure 5H). The ebastine treatment did not exhibit this effect (data not shown), suggesting that UA does not share all of its effects on lipid metabolism with ebastine. Interestingly, our unpublished data suggest that the reduction in the average number of total acyl double bonds of diacylGPLs sensitize cancer cells to CADs (Anand et al., manuscript in preparation).

### 3.6. Sublethal Concentrations of CADs Sensitize MCF7 Cells to UA-Induced LMP and Cell Death 

Prompted by a similar but distinct mechanism of action of CADs and UA on cancer cell lysosomes, we investigated whether the combination of these drugs would result in additive or synergistic cytotoxicity in cancer cells. Indeed, the co-treatment of MCF7 cells with 10 μM of UA and 5 μM of ebastine increased the cell death in comparison to the sum of those induced by either drug alone (Figure 6A). Similar increases in cell death were also observed when UA was added 2 h prior to ebastine, or vice versa (Figure 6A). Pretreatment with UA 2 h prior to ebastine, also had a synergistic effect on LMP as analyzed by a galectin-3 puncta assay (Figure 6C,D), while it did not affect the uptake of ebastine as analyzed by mass spectrometry (Figure 6E). An essentially identical synergy in both the cell death and loss of lysosomal membrane integrity was also observed when UA was combined with a sublethal concentration of another CAD, siramesine (Figure 6B). These data indicate that targeting cancer cell lysosomes simultaneously with two chemically distinct drugs, namely, UA and a CAD, has synergistic anticancer effects and encourages further studies on such combination therapies.

## 4. Discussion

The data presented here illuminates the previously unidentified effects of UA on cancer cells. We provide evidence that UA rapidly and drastically abolishes the functionality of lysosomes in cancer cells. UA neutralizes lysosomes, induces massive LMP, autophagy inhibition, and eventually cell death.

The data showing that UA induces neutralization and massive LMP hours before the induction of MOMP or cell death suggests that these events on lysosomes are not the results of an activated apoptosis pathway. The leakage of the lysosomal contents that follows LMP can lead to an induction of apoptosis through the release of lysosomal cathepsins capable of cleaving and activating the caspases essential for apoptosis induction. Several studies have also shown that cathepsins can be involved in a more direct activation of BAX/BAK-dependent MOMP, which results in apoptosis [60,61]. Moreover, given that Bcl2-overexpressing MCF7 cells were only partially protected against cell death induced by the UA, UA likely induces another, apoptosis-independent cell death pathway such as lysosome-dependent cell death. Lysosome leakage can lead to lysosome-dependent cell death, even though we were not able to provide clear evidence for the role of LMP in UA-induced cell death.

Besides the induction of cell death pathways, disruption of the lysosomal pH and loss of lysosomal integrity have consequences on various cellular processes. The neutralization of lysosomal pH impairs the degrading activity of lysosomes essential for recycling and scavenging, and can cause the formation of toxic digestion products or reactive oxygen species (ROS) [62,63].

The disruption of lysosomal pH and a loss of lysosomal integrity after UA treatment were most likely responsible for the observed depletion of functional autolysosomes capable of quenching eGFP of tfLC3. Whether neutralized lysosomes fuse with autophagosomes is still a subject under debate [64,65]. Lysosomes neutralized by UA may, thus, fail in fusing with autophagosomes or simply produce non-functional autolysosomes with neutral pH. Either way, the UA-induced neutralization of lysosomes results in the impaired degradation of cargos within autophagosomes.

Fusion between autophagosomes and lysosomes is greatly enhanced at the perinuclear region and upon starvation, and lysosomes have been found to cluster in this area to promote autophagy [66,67]. It is, therefore, obvious to speculate that the lysosomal clustering that we observed might have been a compensatory function to upregulate autophagy upon the UA-induced autophagy inhibition.

The present study also demonstrated that UA alters the cellular lipid composition and, thus, lipid metabolism. This effect is likely linked to lysosomal malfunctioning, considering that UA induced the loss of lysosomal hydrolases and shared its traits on cellular lipid composition with the lysosomotropic agent, ebastine. One of the prominent effects of UA on the cellular lipid composition was the increase in the levels of lysoGPLs, lipids with detergent-like properties that can destabilize biological membranes even upon a small increase in levels [68,69]. Saturated lysoGPLs have particularly strong destabilizing effects on membranes [68]. Thus, UA-induced elevation in the saturation levels of diacylGPLs and lysoGPLs may enhance the membrane destabilizing effect of increased lysoGPL levels. We cannot determine whether the UA-induced alterations in the cellular lipid composition contributed to LMP or cell death. Nevertheless, unpublished data from our laboratory demonstrated that increasing the saturation of cellular diacylGPLs sensitizes cancer cells towards CADs, whose cytotoxicity depends on LMP. Interestingly, combining UA and CAD treatments significantly enhanced LMP and cell death compared to either drug alone.

This study also provides a possible explanation for the beneficial effects of UA on multidrug resistance. Several studies have reported that UA re-sensitizes drug-resistant cancer cells towards chemotherapy [14,70,71]. One of the mechanisms by which drug-resistant cancer cells can evade cell death is by storing basic drugs in lysosomes, thus hindering them from reaching their sites of action [72,73,74]. Multidrug resistance is typically reverted by lysosomotropic agents that inhibit lysosomal functioning by disrupting the pH gradient or that cause lysosomal permeabilization [21,75]. Furthermore, autophagy has been suggested to be implicated in the development of drug-resistant cancer cells and autophagy ablation has been shown to revert multidrug resistance [76,77]. Accordingly, it is likely that part of the mechanism by which UA can reverse multidrug resistance is through lysosomal disruption and autophagy inhibition.

Overall, this study demonstrated that UA has a rapid and profound effect on the lysosomes disrupting the pH gradient, altering lysosomal localization, blocking autophagic flux and causing massive LMP before the induction of apoptosis and cell death. Finally, the synergistic anticancer activity of CADs and UA encourage further preclinical studies testing the efficacy of the combined targeting of lysosomes with molecules with distinct physicochemical properties.

## Figures and Tables

**Figure 1 cells-11-04079-f001:**
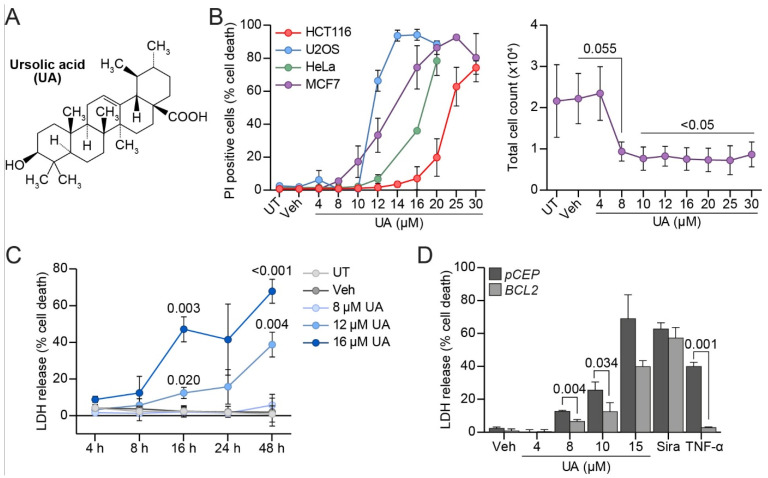
UA halts proliferation and induces cell death partly through apoptosis. (**A**) Structure of UA. (**B**) Cell death (%) in MCF7, HeLa, U2OS, and HCT116 cells after 24 h treatment with indicated concentrations of UA (left). Cell death measured by propidium iodide (PI) uptake assay. Total cell counts of MCF7 cells 48 h after treatment with the indicated concentrations of UA (right). (**C**) Cell death (%) in MCF7 cells after UA treatment. Duration and concentration of treatment indicated on figure. Cell death measured by lactate dehydrogenase (LDH) release assay. (**D**) MCF7-pCEP and MCF7-BCL2 cells treated for 48 h with indicated concentrations of UA, 10 μM siramesine, or 200 ng/mL TNF-α. Cell death measured as in (**C**). Data is shown for a representative experiment. See Suppl. Appendix A for all the independently performed experiments. The *p*-values were defined by a multiple unpaired *t*-test with a Welch’s correction comparing each treatment to the vehicle in (**A**) (see Appendix A for *p*-values), (**B**) (right) and (**C**). A similar test was performed on the triplicates of one experiment comparing the cell death in MCF7-BCL2 to MCF7-pCEP for each treatment in (**D**). Abbreviations: (UT) untreated, (Veh) vehicle.

**Figure 2 cells-11-04079-f002:**
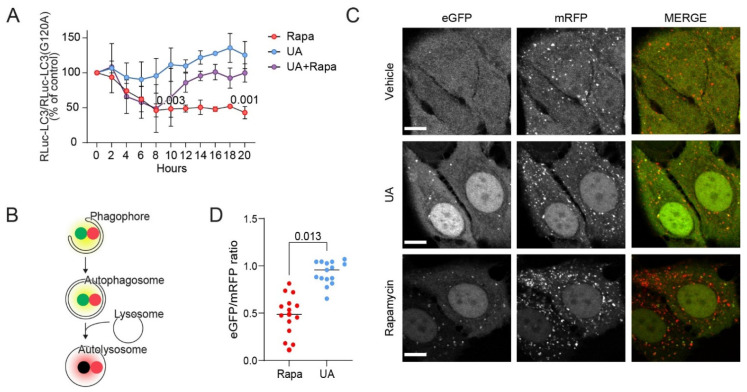
UA inhibits autophagy at the maturation step. (**A**) Autophagic flux assessed by comparing luciferase activity ratios in MCF7-RLuc-LC3 and MCF7-RLuc-LC3(G120A) cells treated with 20 nM of rapamycin (rapa), 8 μM UA or both for the indicated time points. The values were normalized to time 0 (set at 100%). Rapamycin served as the positive control for the autophagy induction [52]. n = 2 for all data points except for 10 h and 20 h where n = 4. (**B**) Illustration of the tfLC3 assay: immature neutral vesicles will express a yellow signal from the eGFP and mRFP coupled to LC3 whereas mature acidic autophagosomes will appear red. (**C**) Representative confocal images of MCF7-tfLC3 cells treated for 24 h with the vehicle (DMSO), 8 μM of UA, or 100 nM of rapamycin. Bars: 10 μm. (**D**) Ratio of eGFP and mRFP signal from the MCF7-tfLC3 cells treated for 24 h with 8 μM of UA or 100 nM of rapamycin. The eGFP/mRFP ratio was quantified for each LC3 punctum. A total of 5 cells were analyzed for each experiment and the mean for each of the 15 cells are shown as data points. The *p*-values were defined by one sample *t*-test comparing each treatment at 10 and 20 h, respectively, to the baseline 100% in (**A**) and an unpaired *t*-test with a Welch’s correction performed on the mean values from the three replicate experiments in (**D**). No statistics were performed on the remaining time points in (**A**) as n = 2 for these. Abbreviations: same abbreviations as in Figure 1.

**Figure 3 cells-11-04079-f003:**
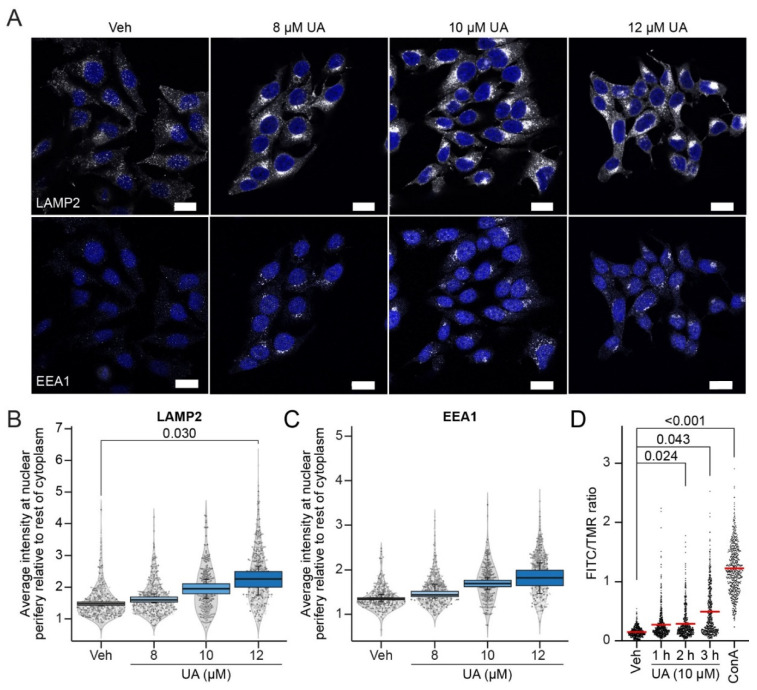
UA alters lysosomal positioning and pH. (**A**) Representative images of MCF7 cells treated with the vehicle (DMSO) or UA at 8 μM, 10 μM or 12 μM for 6 h, and co-stained with the markers of early endosomes (EEA1) and lysosomes (LAMP2). Nuclei were labeled with Hoechst 33342 (blue). Bars: 20 μm. (**B**) SuperPlot of the quantification of the LAMP2 distribution. LAMP2 nuclear proximity was scored as the mean intensity of the LAMP2 signal in a six-pixel wide (~2 µm) ring overlapping with the nuclear periphery over the LAMP2 intensity in the rest of the cytoplasm and plotted as box plots. The scatter and violin plots show the distribution of the individual (per cell) measurements represented by a sample of 150 cells per experiment. Different grey intensities of the points represent the separate experiments. (**C**) Identical to (**B**) but for EEA1 intensities. (**D**) FITC/TMR ratio in MCF7 cells treated with the vehicle (DMSO) or 10 μM UA for 1 h, 2 h, or 3 h. As a positive control, the V-ATPase inhibitor, concanamycin A (ConA, 100 nM), was added for 2 h. The FITC/TMR ratio was measured for each lysosome and each value is represented as a dot. A total of 200 lysosomes per replica were analyzed (with 600 in total per treatment). The red line represents the overall mean. Outliers were identified and removed using an iterative Grubbs test on each replicate with α = 0.0001. The *p*-values were defined in (**B**–**D**) by a multiple unpaired *t*-test with a Welch’s correction on the mean values from the replicate experiments. No statistics were performed on the 10 μM of UA concentration in (**B**,**C**) as n = 2 for this concentration. Each treatment was compared to the vehicle. Abbreviations: same abbreviations as in Figure 1.

**Figure 4 cells-11-04079-f004:**
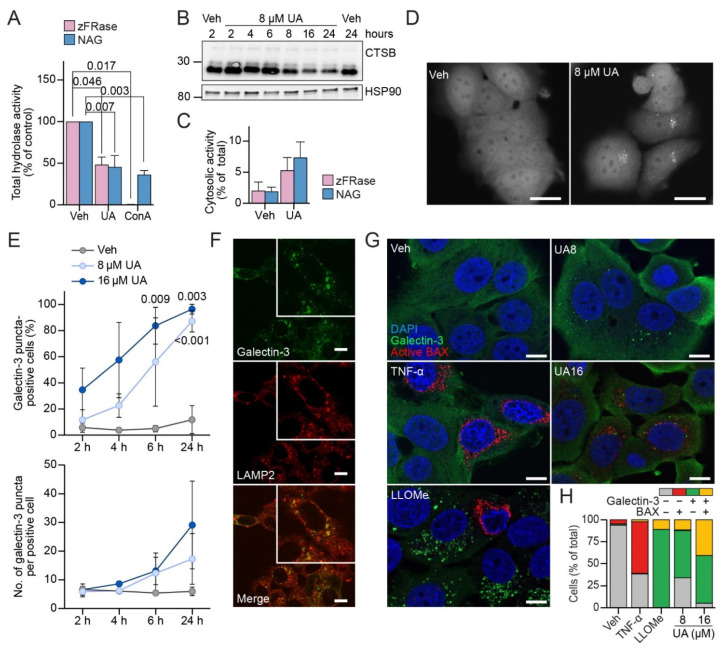
UA induces lysosomal membrane permeabilization before mitochondrial outer membrane permeabilization. (**A**) Total cysteine cathepsin (zFRase) and *N*-acetyl-*β*-glucosaminidase (NAG) activities of MCF7 cells treated with the vehicle (DMSO), 8 μM of UA, or 2 nM of ConA for 24 or 48 h. (**B**) Western blot of mature cathepsin b levels (two bottom bands) and prepro-cathepsin (top band) after treatment with 8 μM of UA or the vehicle (DMSO) for the indicated time points. The sizes of the closest molecular markers are indicated on the left in kDa. Heat shock protein 90 (HSP90) served as the loading control. (**C**) Cytosolic zFRase and NAG activities in MCF7 cells treated with the vehicle (DMSO) or 8 μM of UA for 24 or 48 h. (**D**) Representative images of MCF7-eGFP-LGALS3 treated as indicated for 24 h. Bars: 20 μm. (**E**) MCF7-eGFP-LGALS3 cells treated as indicated and the percentage of cells with ≥3 galectin-3 puncta was quantified along with the number of galectin-3 puncta per galectin-3 puncta-positive cell. A total of more than 1000 cells/sample were quantified. (**F**) Representative confocal images of MCF7 cells treated with 16 μM of UA for 24 h and stained for galectin-3 (green) and LAMP2 (red). Bars: 10 μm. Close-up illustrates the colocalization. (**G**) MCF7 cells treated with 8 or 16 μM of UA or 1.5 mM of LLOMe for 6 h or with 20 ng/mL of TNF-α for 24 h and co-stained for galectin-3 (green) and active BAX (red). Nuclei were labeled with Hoechst 33342 (blue). Bars: 10 μm. (**H**) Quantification of images in (**G**) by the manual counting of five randomly selected fields (50–100 cells). For a more detailed representation, see Appendix A. The *p*-values were defined by a multiple unpaired *t*-test with a Welch’s correction before normalizing to the control in (**A**). Each treatment was compared to the vehicle. The *p*-values were defined by a multiple unpaired *t*-test with a Welch’s correction comparing each treatment to the vehicle in (**C**,**E**). Abbreviations: (HSP90) heat shock protein 90, otherwise the abbreviations are the same as in Figure 1.

**Figure 5 cells-11-04079-f005:**
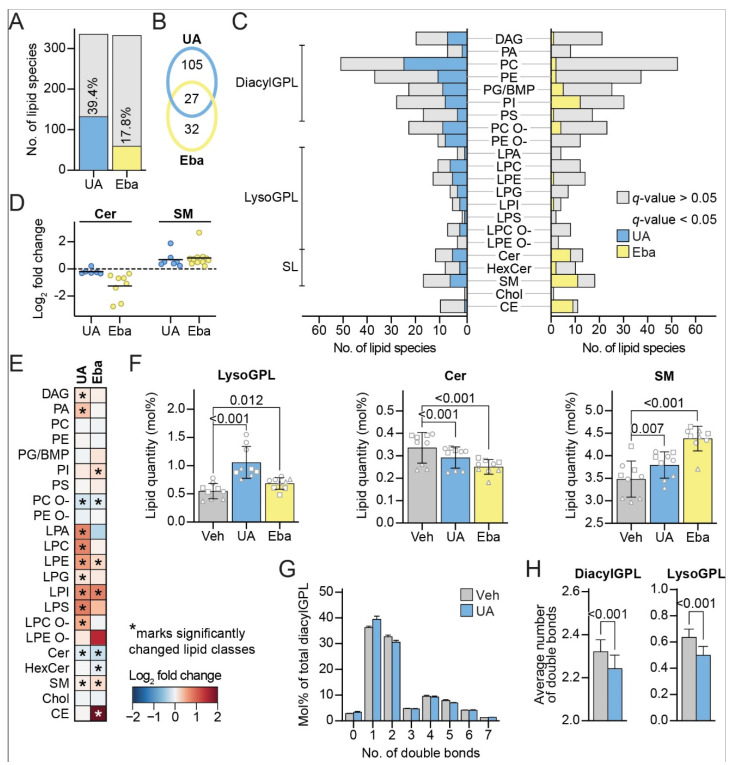
UA shifts the lipid profile towards membrane instability. Lipidomics analysis of cells treated with the vehicle (DMSO) for 8 h, 8 μM of UA for 8 h or 5 μM of ebastine for 6 h. (**A**) Total number of lipid species included in the analysis in UA vs. the vehicle (335) and ebastine vs. the vehicle (332) comparisons (light grey). Percentage of detected lipid species that were significantly changed after treatment with either 8 μM of UA for 8 h (blue) or 5 μM of ebastine for 6 h (yellow) relative to the vehicle (DMSO)-treated cells. (**B**) Venn diagram showing the numbers of statistically significantly altered lipid species unique or shared between the UA and ebastine treatments. (**C**) Bar graph showing monitored lipid species in the indicated class (grey) and number of significantly changed lipids upon treatment vs. the control. (**D**) Significantly changed ceramide species shown as a log2 fold change of UA and ebastine-treated cells compared to the vehicle (DMSO) (left), and the same for sphingomyelin (right). (**E**) Heatmap showing log2-transformed fold changes of the different lipid classes in the UA- and ebastine-treated cells compared to the vehicle (DMSO). Classes that were significantly changed (*q*-value < 0.05) are marked with an asterisk. (**F**) Levels of lysoGPL (left), ceramide (middle) and sphingomyelin (right) given in mol% after treatment with the vehicle (DMSO), UA or ebastine. Same shaped data points represent replicates of the same experiment to illustrate day-to-day variation. (**G**) Distribution of lipid species in relation to double bonds given in percentage of the total diacylGPL category. (**H**) Average number of double bonds per single lipid in the diacylGPL (left) and lysoGPL (right) categories. Linear modelling with a Benjamini–Hochberg correction was used to determine the significantly changed lipid species in (**A**–**D**), significantly changed lipid categories and classes in (**E**,**F**) and the change in average double bond per single lipid species in (**H**). Linear modelling was performed on triplicates from three independent experiments (nine data points in total) accounting for the batch factor. Abbreviations: (eba) ebastine, (BMP) bis(monoacylglycero)phosphate, (GPL) glycerophospholipid, (SL) sphingolipid, (DAG) diacylglycerol, (PA) phosphatidic acid, (PC) phosphatidylcholine, (PE) phosphatidylethanolamine, (PG) phosphatidylglycerol, (PI) phosphatidylinositol, (PS) phosphatidylserine, (PC O-) acyl-alkyl PC, (PE O-) acyl-alkyl PE, (LPA) lysoPA, (LPE) lysoPE, (LPC) lysoPC, (LPG) lysoPG, (LPI) lysoPI, (LPS) lysoPS, LPC O-acyl-alkyl LPC, (LPE O-) acyl-alkyl LPE, (Cer) ceramide, (HexCer) hexosylceramide (SM) sphingomyelin, (Chol) cholesterol, and (CE) cholesteryl ester. Otherwise, abbreviations are the same as in Figure 1.

**Figure 6 cells-11-04079-f006:**
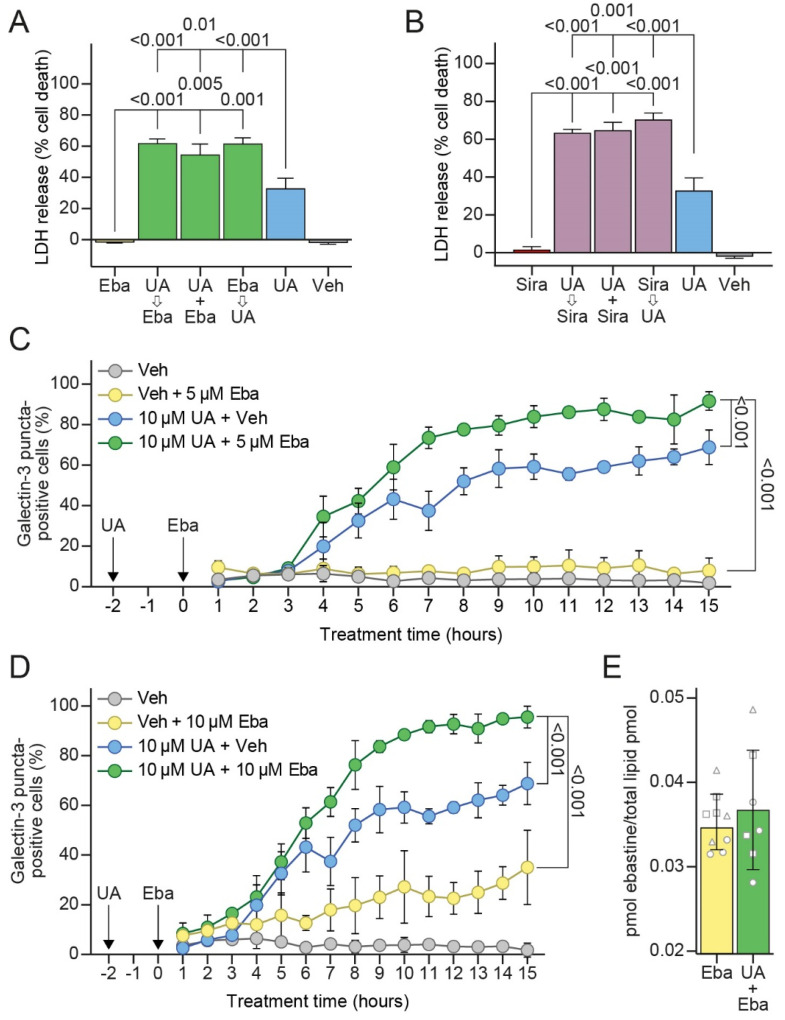
UA and CAD combination treatment enhances cell death and LMP. (**A**) Cell death measured in MCF7 cells pre-treated with the vehicle (DMSO), 10 μM of UA or 5 μM of ebastine 2 h prior to treating with 10 μM of UA and/or 5 μM of ebastine for an additional 48 h. Cell death measured as in Figure 1C. (**B**) Same as (**A**) but for 3 μM of siramesine. (**C**) Galectin-3 puncta formation in MCF7-eGFP-LGALS3 cells pre-treated with the vehicle (DMSO) or 10 μM of UA for 2 h and subsequently treated with the vehicle (DMSO) or 5 μM of ebastine for up to an additional 15 h. The percentage of cells with ≥3 galectin-3 puncta was quantified. (**D**) Same as (**C**) for ebastine 10 μM. (**E**) MCF7 cells pre-treated for 2 h with the vehicle (DMSO) or 8 μM of UA and subsequently treated with 5 μM of ebastine for an additional 6 h. Ebastine levels were measured using mass spectrometry and normalized to the total lipid content. Same shaped data points represent replicates of the same experiment to illustrate day-to-day variation. The *p*-values were defined by a multiple unpaired *t*-test with a Welch’s correction comparing each combination treatment to each single treatment in (**A**,**B**), an unpaired *t*-test of the area under the curve (AUC) values with a Welch’s correction comparing the combination treatment to each single treatment in (**C**,**D**), and an unpaired *t*-test with a Welch’s correction in (**E**). Abbreviations: same abbreviations as in Figure 1 and Figure 5.

## Data Availability

Data is available upon request.

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
