# Peer review of "Ursolic Acid Impairs Cellular Lipid Homeostasis and Lysosomal Membrane Integrity in Breast Carcinoma Cells"

_cells, 2022, doi:10.3390/cells11244079_

Round 1
Reviewer 1 Report
The authors, in the current study, aim to propose a mechanism for ursolic acid in inducing apoptosis in cancer. They have identified lysosomes as essential targets of the anti-cancer activity of ursolic acid. They propose that the disruption of lysosomal function by the molecule impairs the autophagic pathway. I have the following comments:
1. The authors state that they have identified lysosomes to be an essential target for anti-cancer activity of UA which is mediated by apoptosis. However, they fail to experimentally show that activity of UA on lysosomes is essential for it to induce apoptosis. Following experiments may be performed to unequivocally support this claim:
a. Effect of autophagy induction and inhibition on UA induced apoptosis
b. Effect of knocking down essential autophagy genes like LC3 and LAMP2A on UA induced apoptosis
2. Table S3 consists of the list of lipid groups identified. More information with the lipid subclasses identified (the types of fatty acid chains present) along with their characteristic ions in MS1 and MS2 may be included.
3. Significance levels should be mentioned for fig 1B
Reviewer 2 Report
This review is for the manuscript titled "Ursolic acid impairs cellular lipid ..." by DL Fogde, et al. This investigation comes from the laboratory of Dr. M. Jaattela who is a highly regarded researcher in the field. This investigation is represents a large body of work, is well organized, carefully conducted, and clearly written. Nevertheless, there are some minor issues that need to be addressed prior to publication.
1. The text, images and legend of Figure 2 need clarification. Fig 2A and Supplemental Figs S2A & B - LC3 punctate formation and inhibition of autophagic flux in MCF7 cells seem contradictory - please clarify.
Fig 2C - Images of Rapa treated cells are not explained well in the text and as illustrated in 2B. The text of "roughly equal numbers of red and yellow puncta" does not support this. Should not there be more red punctate here?
Fig 2D - No data is shown for vehicle as noted in the legend. Line 337 legend should be "in (A)". Line 338 should be "in(D)".
2. Figure 5A - It is unclear how the number of altered lipid species stated in the text were obtained. For UA, 39.4% of 298 = 117, not 139 noted in the text. For Eba, 17.8% of 295 = 53, not 59 as noted in the text. What is this review missing? It should be noted that lipidomics is out of the expertise of this reviewer.
3. Line 412 - The use of z-FRase for total lysosoma cysteine proteases is not completely accurate. It is used for cathepsin B and L (see PMID 22464443, 33436436).
4. Line 630-631. Lysosome storage is not the only mechansim of drug resistance as stated in the text. The authors should revise the text.
5. The legend in Figure 3 notes replicates a few times which raises the question of independent experiments. Is it N=3?
Very Minor
6. Line 90 - Table 1 should be Table S1.
7. Line 275 - In legend of Figure 1 Figure 1C should be Figure 1B
8. Lines 441-442 - text should be "...zFRase and NAG activity"
9. Line 478 - "phospholipidosis"
10. Line 510 - "extent"
11. Legends of Figs 5 & 6 - The phrase "data points are shaped according to replicates" is unclear to this reviewer. Are these all replicates from all experiments? Replicates from one experiment? Why are they shown?
Round 2
Reviewer 1 Report
The authors have addressed the concerns raised by me. The manuscript may be accepted in its current form.